# Identification of Variables That Cause Agricultural Graduates Not to Return to the Rural Sector in Ecuador. Application of Fuzzy Cognitive Maps

**Bélgica Bermeo Córdova [1,2,*] , José Luis Yagüe Blanco [1] , Maritza Satama [3,4] and Carlos Jara [5]**

1   Department of Agroforestry Engineering, Universidad Politécnica de Madrid, 28040 Madrid, Spain; joseluis.yague@upm.es
2   Nutrition and Community Health, Universidad Técnica del Norte, Ibarra 100105, Ecuador
3   Department of Agricultural Economics, Universidad Politécnica de Madrid, 28040 Madrid, Spain; maritza.satama.bermeo@alumnos.upm.es
4   CEIGRAM, Universidad Politécnica de Madrid, Calle Senda del Rey 13, 28040 Madrid, Spain
5   Department of Agriculture, Universidad Central del Ecuador, Quito 170520, Ecuador; carlosjulio.jara@gmail.com
*   Correspondence: bn.bermeo@alumnos.upm.es

**Abstract:** In Ecuador, the factors that encourage agricultural graduates to return to the rural production sector are unknown. Therefore, this study seeks to identify the variables that influence why agricultural graduates do not return to the rural sector. Fuzzy cognitive maps and hierarchical clustering were used to achieve this. Interviews were carried out with academic experts from the rural development field and agricultural planning managers who work across 10 of Ecuador's provinces. Eighteen categorized variables were identified in the study, of which five were strategies for agricultural development that involve different actors. These were: (1) State policies for rural development, (2) quality academic training, (3) innovation and entrepreneurship training, (4) national research and university extension programs, and (5) associative cooperativism. The identification of these variables supports state authorities and universities in designing strategies that promote the involvement of agricultural graduates in the rural production sector.

**Keywords:** agricultural graduates; quality academic training; state policies; rural development; fuzzy cognitive maps; Ecuador

## 1. Introduction

Graduates not returning to where they came from, as well as the migration of the rural population and concentration of land ownership amongst corporations, all contribute to the scarcity of the workforce in the world's agricultural sectors [1–3]. This signals the need to establish structural social, economic, political, and environmental adjustments to encourage professionals to remain in the rural sector. Niedomysl and Amcoff [4] argue that qualified people and those who have managerial roles are less likely to return to where they came from. However, Du [5] states that younger family members with purchasing power are more likely to return after their graduation, compared to those who, due to their family's economic situation, studied within their province and subsequently leave the area they came from. This means that contextual factors influence whether people who leave the countryside and migrate to the big cities return. For example, in Switzerland, Rérat [1] determines that graduates not returning depends on factors such as family relationships, source of livelihood, and the labor market; factors which can vary amongst the graduates. However, the labor market becomes the primary factor that affects whether they return to where they came from.

The dynamic of the migration of young people from rural areas who go to study at university is common across the world. However, the post-graduation behavior is different [6]. In developed countries such as the United States, graduates return to where they came from to work in the public and private sectors, and very few return to work autonomously [7]. In research based on a case study in Italy relating to the quality of education and public support for the internal migration of graduates, Ciriaci [8] highlights that universities are responsible for producing a competitive, young workforce for local companies. Similarly, the government is responsible for improving the employability of qualified workers and encouraging graduates to return to the rural sector.

As a result, the higher education system is aimed at creating and disseminating knowledge, focused on the development of new processes and technologies, to improve productivity through innovation and product development, leading to economic growth. However, a university should move towards social development [9], for which it could opt to use the link between higher education institutions and companies as a strategy for academic training, by using internships or placements [3]. The United Nations [10] states that coordination between public and private actors and social organizations helps to achieve sustainable development goals in the environmental, economic, and social fields; where the university should be a partner for the interested parties and a promoter of territorial development [11]. While these declarations are important for achieving academic training with a social vision; the global geopolitics of the university phenomenon, according to Brunner [12], leads to the segmentation of universities based on power, wealth, and prestige. This has led to universities, especially in Latin America, following a hegemonic model and adopting processes that do not emphasize the social development of the majority.

On the other hand, in the rural sector, the problem facing the agricultural sector has been addressed by the public sector, the cooperation bodies, and academia, taking into account their areas and disciplines, without being able to connect with topics such as production structure, policies, environment, or rural economies, amongst others [13]. In Mexico, for example, the lack of arable land, access to credit, and lack of support from communities, local institutions, and organizations for working in rural areas, means that youngsters have little interest in rural development [14]. A similar situation is observed in Ecuador, where the rural predicament is demonstrated by land that has been destroyed, eroded, with distribution problems and government policies that barely dignify rural production, if at all [15]. The lack of public and private investment relates to a severe economic crisis, in addition to automation processes that displace the workforce. This creates problems in the labor market, in particular for women who are carrying out low value work. Therefore, there are few incentives for building areas with diversified production and investing in creativity to ensure long-term economic development [16,17].

In Ecuador, Estévez [18] reported that in 2017, of the 16.4 mil people who lived in rural areas, only 1.5 mil were young people between the ages of 15 and 29. According to information from various studies, young people did not want to live in rural areas, not only due to the lack of employment opportunities, they also cite living conditions, low levels of productivity, lack of entertainment, depletion of natural resources, communication channels, etc. [19,20]. This implies that young people feel that the rural environment does not offer them a viable future [21]. Studies carried out in Ecuador determine that approximately 60% of farmers are between 46 and 75 years old, which means that the majority of farmers are not youngsters; in the countryside, 8 out of every 10 adults live in poverty [22]. According to the Employment, Unemployment, and Underemployment Survey [23] in Castilla de León-Spain, of an economically active rural population made up of 2,764,876 people, 20.2% are employed with a basic monthly salary of 394 dollars (as determined by law), 77.6% are underemployed and do not have a regular monthly income, whilst 2.2% are unemployed, without any kind of income. These figures highlight the need to increase employment opportunities in rural areas to counteract the migration from the countryside to the cities, which can lead to human capital flight from rural communities. As a result, in considering the rural production sector and the activity carried out by the agricultural science professionals, it is necessary for their training to be aimed at leading agricultural education and training processes that drive production, exports, and food sovereignty [24].

In that regard, Ecuador has not been able to overcome its farmers thinking that they are only small producers who supply food for the national population. Data relating to the period 2013–2017 obtained from the Ministry of Higher Education, Science, Technology & Innovation (SENESCYT) showed that farming, agriculture, livestock, and zootechnics degrees in Ecuadorian universities have an average annual growth rate of −2.1%. This means that there is a declining population of agricultural graduates in the country. Ponce [25] states that the overall increase in migration of rural technical skills in Ecuador could hypothetically be linked to: Firstly, the poor quality of academic training and the links that students establish with rural society and its surrounding communities, resulting in low levels of research and innovation, as well as knowledge and technology transfer based on the sector's needs. Secondly, weak agricultural policies of an inclusive nature that respect food sovereignty and environmental sustainability, considering the real needs of small and medium rural producers. Thirdly, low levels of associativity that reduce the negotiation power in the production chain, resulting in intermediation. Finally, poor conditions for promoting entrepreneurship aimed at diversifying production and creating added value.

Paradoxically, the educational coverage in Ecuador has been improving, but the teaching methodologies have few practical connections that enable them to solve problems in the rural sector [26]. There is good educational infrastructure, but the results are weak in terms of rural development driven by knowledge. In this context, academia faces the challenge of training agricultural professionals in critical thinking, with knowledge of the social, economic, and environmental context. To ensure that the agricultural graduates have the ability to work in an inter and transdisciplinary manner, with technical-scientific knowledge acquired from being part of a formal group [13]. This requires academic training to be part of participatory processes, which means going beyond the theory and acquiring practical experience and endogenous knowledge from an external sector, where it can interact with social sectors, local and national bodies, non-governmental bodies, and the production sector [27].

Knowledge of the new rural environment in Ecuadorian universities and the redefinition of the Ecuadorian rural predicament are falling behind, but there are situations such as the case of the Santiago de Guayaquil Catholic University, where the students created a map for two rural areas showing agricultural education schools and their perspectives of working in these areas, culminating in the identification of existing needs and opportunities [28]. This demonstrated the significant contribution that academia can have at a local level, taking into account its interdisciplinary structure. The contributions can include the design of public policies, providing information to those responsible for creating the policies. If information is a public service, the state must fund these projects, and academia should involve students, fulfilling the legacy of learning-by-doing. As a result, this study provides an understanding as to whether the state's policies aimed at the rural sector, quality academic training, and entrepreneurship are factors that determine whether agricultural graduates return to the rural production sector. To do so, the study aims to understand the experts' perceptions, with the aim of identifying the variables that affect the behavior of agricultural graduates who do not return to the rural sector after completing their studies. This enables general, but useful, answers to be obtained with regards to changing university policies.

## 2. Materials and Methods

The study was carried out based on interviews targeted at teachers of agricultural degrees and professionals with experience in rural development and farming in Ecuador. This was done in three consecutive stages: (i) Design and validation of the interview questionnaire, (ii) carrying out the interviews, and (iii) systematization and analysis of information.

### 2.1. Design and Validation of the Interview Questionnaire

In order to understand the variables that influence the relationship between the agricultural graduates and the rural sector in Ecuador, a semi-structured interview script was designed. The following aspects were taken into account when designing the script:

(a)	Agricultural graduates' motivation for being involved in agricultural development,

(b)	employment opportunities for agricultural professionals in the rural sector,

(c)	generational transfer of knowledge and experience that exists in the rural sector,

(d)	the relevant actors who work in rural production development,

(e)	strategies for strengthening the relationship and work between academia and the agricultural sector,

(f)	academic training for agricultural development.

The script was validated by five university professors linked to various agricultural faculties in Ecuador.

### 2.2. Collection of Information

In order to understand the opinions on the research topic, a sample was taken of 38 rural development experts in Ecuador, selected by using convenience sampling and the snowball technique [29]. The information was obtained through both conventional and virtual in-depth semi-structured interviews between 2018 and 2019. The opinions given by the 38 interviewees were sufficient for obtaining the most relevant information [30,31]. The interviews were divided in two research groups: Academic and agricultural planning sector. The main purpose to establish two categories was to analyze perspectives from academic vision, and an integrated management vision. Fifteen interviews were carried out with university agricultural lecturers, and this formed the first research group (University Lecturers—ULs). The 23 remaining interviews formed part of the second research group (Agricultural Planning Managers—APMs) and were aimed at: 10 in the public sector, 4 Ecuadorian NGOs, 1 international NGO in Ecuador, 3 representatives from the private sector, and 5 rural development consultants. The 38 interviewees work in 7 different planning zones [32] across 10 of Ecuador's 24 provinces.

The experts who were interviewed had an average age of 53 ± 7. In this research population, 65.8% have an agricultural or farming Bachelor degree, while 34.2% are in the agricultural industry or natural resource sector. It is important to mention that 62.3% of all the experts interviewed had a postgraduate degree in rural development. The level of academic training and field of education in agriculture of the group of experts interviewed ensured the quality of the sample and information in the research [33].

### 2.3. Systematization and Analysis of Information

Following the methodology presented by Hernández et al. [34], the information from the 38 interview recordings was transcribed individually in Word format and analyzed in two phases: In the first phase, the concepts were identified based on the transcription obtained using Excel worksheets, taking into account the six topics from the interview script. A total of 117 variables were obtained (Appendix B). In the second phase, the variables were categorized based on the similarity of the topics, and then causal relationships were established. As a result, 18 categorized variables were obtained, which based on the perceptions of those interviewed, were organized into: Problems and strategies, with the responsibility of being served by academia, the state, and the rural sector itself (cooperation, private companies, and the population who live there), as shown in Table 1 (see Appendix A for the definition of categorized variables). It is essential to mention the categorization process was discussed among authors to establish a collective agreement from different branches of expertise.

**Table 1.** Categorized variables obtained from the people who were interviewed.

| Categorized Variables | | Responsible Actors | | |
|---|---|---|---|---|
| **Initial Format** | **Condensed Format** | **Academia** | **State** | **Rural Sector** |
| 1. State policies for the development of the rural sector | State policies for rural development | Strategy | Strategy | Strategy |
| 2. Basic services that are lacking in the rural sector | Basic services | | Problem | Problem |
| 3. Problems with commercializing agricultural products | Commercialization | | | Problem |
| 4. Inadequate practical training of agricultural professionals | Practical training | Problem | | Problem |
| 5. Low levels of associative cooperativism between farmers | Associative cooperativism | | | Strategy |
| 6. Inadequate strategic, tactical, and operational planning in the rural farming sector | Strategic, tactical, and operational planning | | Problem | Problem |
| 7. Insufficient subsidies for farmers in the rural sector | State subsidies | | Problem | |
| 8. Insufficient business and commercial development in the rural sector | Business and commercial development | | | Problem |
| 9. Diversification of activities in the rural sector | Diversification of activities | | | Strategy |
| 10. Quality academic training | Quality academic training | Strategy | Strategy | |
| 11. Risk in the farming sector | Risk in the farming sector | | | Problem |
| 12. Difficulty accessing credit | Credit | | Problem | |
| 13. Redesign of the university agricultural curriculum | Curriculum redesign | Strategy | | |
| 14. Agricultural professionals trained to secure a job rather than start a business | Find employment | Problem | | |
| 15. Promote innovation and entrepreneurship in academic training | Innovation and entrepreneurship training | Strategy | | |
| 16. Reorganization of legislated pre-professional practical training in the rural sector | Legislated pre-professional practical training | Strategy | Strategy | |
| 17. Create a national research and university extension program | National research and university extension program | Strategy | Strategy | |
| 18. Insufficient availability of appropriate technology in the rural sector | Appropriate technology for the rural sector | | | Problem |

* Rural Sector = Cooperation, private company and population.

*2.4. Analysis of the Information*

The information was analyzed using Fuzzy Cognitive Maps (FCM) and Hierarchical Clustering.

2.4.1. Fuzzy Cognitive Maps

Fuzzy Cognitive Maps (FCM) are part of the modelling techniques for complex systems and show the cause and effect relationships [35]. In the study, they were used to analyze the rural development experts' individual and group perceptions. FCM enable semi-quantitative studies to be carried out [36] using text analysis [37,38]. The frequency of each variable was determined by taking into account the number of times the interviewee referenced the variable in the guideline questions. The responses were represented using variables that connect one another [35]. Each connection between two variables (cause–effect) is represented by a direction and a weight. A weight higher than zero indicates positive causality, and a weight equal to zero indicates that there is no relationship between the two variables. The strength of the connections between variables is measured on a scale of values [−1,1] [39,40].

Combining the data obtained from various experts in the field, with many different opinions on the topic being studied, improves decision making and reduces the possibility of errors [41]. In this study, the FCM's adjacency matrices (square matrix) were used to represent the weights of the connections between two variables [39,40]. The adjacency matrices for each interviewee were obtained by using the cause–effect relationship between the variable being analyzed and the remaining variables. The normalization of the matrices was obtained by dividing each value content in the matrix by the highest value identified in the matrix. Individual matrices were created, as well as matrices for each research group and the community matrix, which corresponds to the 38 interviewees. The total values (*A*) were divided by the number of interviewees (*K*) in each research group, represented in the Equation (1):

$$A = \frac{1}{k}(A1 + A2 + \ldots + A_k) \tag{1}$$

The indices obtained from the FCM were established using the method stated by Steven et al. [40] and Özesmi and Özesmi [42]. FCMapper Software [43] was used, and the weights of the 18 categorized variables that were established in the adjacency matrices were included. Indices were formed individually, by research group and for all of the 38 interviewees, with these being: (i) Outdegree, the sum of the rows of absolute values of coefficients from the adjacency matrix associated with the connectors that leave the variable. The high outdegree values influence the other variables and are considered driving forces, (ii) indegree, the sum of the columns of absolute values of coefficients from the adjacency matrix associated with the connections that enter the variable. The high indegree variables are considered response variables, and (iii) centrality, the sum of the outdegree and indegree indices, which indicates the variable's level of participation or importance.

Furthermore, as established by Özesmi and Özesmi [42], in order to understand the participation level of the variables within the system, the following were determined: (i) Transmitter variables (forces/drivers), send connections with positive weights towards the other variables, (ii) receiver variables (responses, results/final outcomes), receive connections with positive weights from other variables, (iii) ordinary variables (factors/means), connections enter and leave with not null weights, and (iv) the complexity index, the relationship between the receiver and transmitter variables, whose values represent complex thinking systems. It is essential to consider the number of transmitter and receiver variables, which give us information about what is happening in the FCM. Thus, a high number of receiver variables involved outputs as system results, while a large number of transmitter variables mean a low level of causal arguments in the map [44]. The density index represents the connectivity level from the causal-relationships between the variables in the map. The index presents values between 0 and 1. A high value of this index indicates a greater number of receiver variables than transmitter variables in the system [45]. Pearson's method was applied to the indices to establish whether there were similarities (similar themes) between the 18 categorized variables in the research groups.

### 2.4.2. Hierarchical Cluster

As the study's objective is to identify the variables that influence whether agricultural graduates return to the rural sector, Hierarchical Cluster analysis was carried out [46] using centrality values for the 18 categorized variables obtained in the overall FCM. The IBM SPSS Statistic 25 program was used to identify the groups that were similar (cluster), taking into account the Euclidean squared distance and Ward's clustering method, as the idea is to minimize the variance between the hierarchical cluster and maximize the homogeneity within the groups [47]. The ANOVA calculation was carried out to evaluate the significant differences between the two groups. Descriptive statistical analysis was also carried out based on the mean and standard deviation that was calculated for each cluster.

## 3. Results

### 3.1. Categorized Variables

The categorized variables, which the analysis of information obtained in the study was based on, as well as the repetition of each variable established in the interview for each research group, are detailed in Table 2.

The four most frequent variables reflect the most frequently mentioned categorized variables in both research groups and are related to: "State policies for rural development" (academia/state/rural sector) and "Quality academic training" (academia/state). It is clear that both research groups agree on the importance of these themes, which involve academia, the state, and the rural sector; however, they do not agree on the exact level of importance, which suggests that the groups have different priorities.

**Table 2.** Categorized variables and frequency established in the interviews, split by group.

| Categorized Variables | Frequency | |
|---|---|---|
| | (ULs) | (APMs) |
| State policies for rural development | 34 | 40 |
| Quality academic training | 29 | 43 |
| Innovation and entrepreneurship training | 25 | 31 |
| Risk in the farming sector | 20 | 12 |
| Associative cooperativism | 18 | 20 |
| Find employment | 16 | 14 |
| National research and university extension program | 15 | 16 |
| Curriculum redesign | 13 | 0 |
| Credit | 9 | 15 |
| Business and commercial development | 0 | 17 |
| Commercialization | 5 | 10 |
| State subsidies | 11 | 7 |
| Strategic, tactical and operational planning | 6 | 4 |
| Basic services | 5 | 8 |
| Appropriate technology for the rural sector | 0 | 8 |
| Practical training | 0 | 7 |
| Legislated pre-professional practical training | 6 | 0 |
| Diversification of activities | 0 | 7 |

**ULs** = University lecturers; **APMs** = Agricultural Planning Managers.

### 3.2. Fuzzy Cognitive Map Indices (FCM)—Individual and by Research Group

The individual and group FCM indices are shown in Table 3. The calculation of the variance between the two research groups provided a Fisher value of 0.743 ($p = 0.538$). This means that the two research groups (ULs and APMs) do not show statistically significant differences between the categorized variables indices.

**Table 3. Fuzzy Cognitive Map** (FCM) indices by research group (ULs & APMs).

| Index | FCM Average Values X ± SD | | Values Grouped in FCM | | |
|---|---|---|---|---|---|
| | ULs | APMs | ULs | APMs | ULs-APMs |
| Number of maps | 15 | 23 | 1 | 1 | 1 |
| Number of variables (N) | 6.40 ± 1.12 | 6.30 ± 1.72 | 14 | 16 | 18 |
| Number of connections (C) | 11.13 ± 3.94 | 9.83 ± 4.13 | 68 | 76 | 108 |
| Connections by component (C/N) | 1.72 ± 0.43 | 1.52 ± 0.34 | 4.86 | 4.75 | 6 |
| Number of transmitter variables (T) | 1.33 ± 0.49 | 1.35 ± 0.49 | 1 | 1 | 1 |
| Number of receiver variables (R) | 2.20 ± 0.77 | 1.91 ± 0.79 | 3 | 2 | 2 |
| Number of ordinary variables | 2.87 ± 1.19 | 3.04 ± 1.33 | 10 | 13 | 15 |
| Level of complexity (R/T) | 1.77 ± 0.75 | 1.52 ± 0.73 | 3 | 1 | 2 |
| Density | 0.33 ± 0.10 | 0.31 ± 0.11 | 0.35 | 0.32 | 0.33 |

**X** = Median; **SD** = Standard deviation; **ULs** = University Lecturers; **APMs** = Agricultural Planning Managers; **ULs-APMs** = Community group.

The transmitter variable "National research and university extension program" (academia/state) and the receiver variable "Commercialization" (rural sector) that are shown in Table 4, are common in both of the research groups and the community group (both research groups), which shows the priority of these variables.

**Table 4.** Transmitter, receiver, and ordinary variables for the FCMs in the two research groups and community group.

| Categorized Variables | Research Groups | | Community Group |
|---|---|---|---|
| | (ULs) | (APMs) | |
| State policies for rural development | ordinary | ordinary | ordinary |
| Basic services | receiver | ordinary | ordinary |
| Commercialization | receiver | receiver | receiver |
| Practical training | - | ordinary | ordinary |
| Associative cooperativism | ordinary | ordinary | ordinary |
| Strategic, tactical, and operational planning | ordinary | ordinary | ordinary |
| State subsidies | ordinary | ordinary | ordinary |
| Business and commercial development | - | ordinary | ordinary |
| Diversification of activities | - | ordinary | ordinary |
| Quality academic training | ordinary | ordinary | ordinary |
| Risk in the farming sector | ordinary | ordinary | ordinary |
| Credit | receiver | ordinary | ordinary |
| Curriculum redesign | ordinary | - | ordinary |
| Find employment | ordinary | ordinary | ordinary |
| Innovation and entrepreneurship training | ordinary | ordinary | ordinary |
| Legislated pre-professional practical training | ordinary | - | ordinary |
| National research and university extension program | driver | driver | driver |
| Appropriate technology for the rural sector | - | receiver | receiver |

**ULs** = University lecturers; **APMs** = Agricultural Planning Managers.

### 3.3. Centrality Values for the Research Groups' (ULs & APMs) Fuzzy Cognitive Maps

The 18 categorized variables are reflected in the radial diagram in Figure 1. The ULs and APMs research groups were in agreement on 14 of the variables, of which five had values higher than 1.00 (1.00–2.62) and nine variables had values lower than 1.00 (0.23–0.98) (see Appendix C). The five categorized variables with the highest scores were: (1) "State policies for rural development" (academia/state/rural sector), (2) "Quality academic training" (academia/state), (3) "Innovation and entrepreneurship training" (academia), (4) "National research and university extension program" (academia/state), and (5) "Associative cooperativism" (rural sector). Although the two groups (ULs and APMs) agreed on the importance of these variables (areas), they did not agree on the priority of each of these. For instance, the ULs

research group in the variable "Quality academic training" presented a centrality value of 1.91, while APMs research group reported a greater centrality value of 2.28.

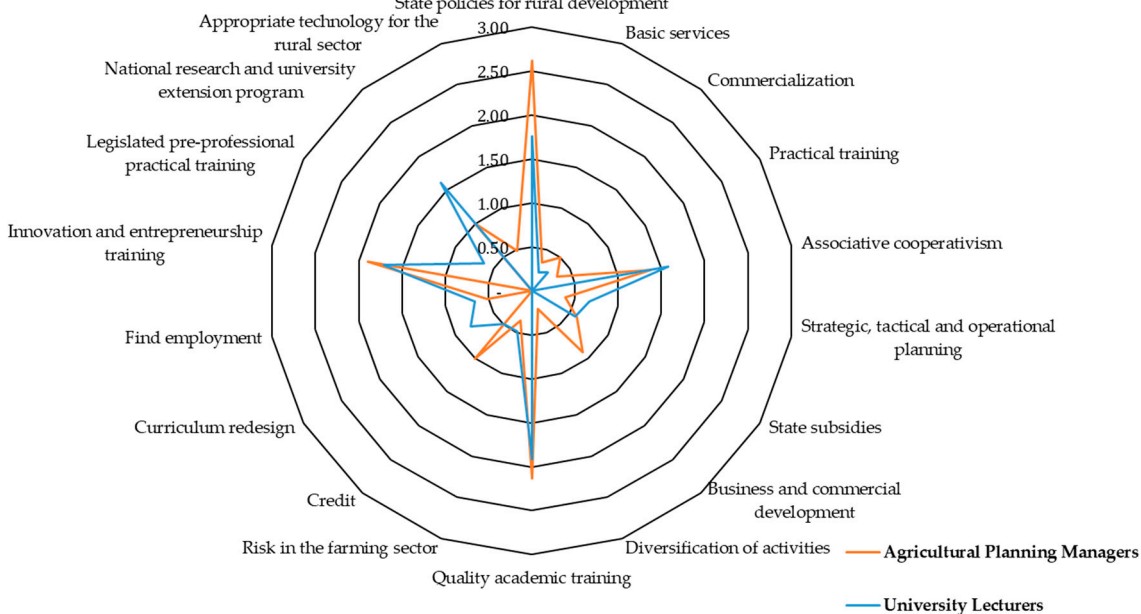

**Figure 1.** Centrality values for the University Lecturers and Agricultural Planning Managers research groups, represented in a radial diagram.

*3.4. Aggregated Community FCM Map for the 18 Categorized Variables for the Two Research Groups*

The centrality values for the categorized variables in the community group (ULs and APMs together) that are shown in Appendix C are reflected by the five largest circles shown in Figure 2. The positive and negative connections between variables are represented by the lines.

*3.5. Hierarchical Cluster with Centrality Values for the Community FCM*

The centrality values of the aggregated FCM classified the 18 Categorized Variables in two clusters, as shown in the dendogram in Figure 3. Cluster 1 groups the five variables that are most similar and relevant, classified as "Strategies for rural development". Whilst cluster 2 groups the remaining 13 variables, classifying them as "Academia and rural problems". Taking into account the centrality of the variables represented in the dendogram, highly significant differences were found between the two clusters (with a value of $p = 4.03\text{e-}08$ with $\alpha = 0.05$), which means within each group presents similar profiles in their values.

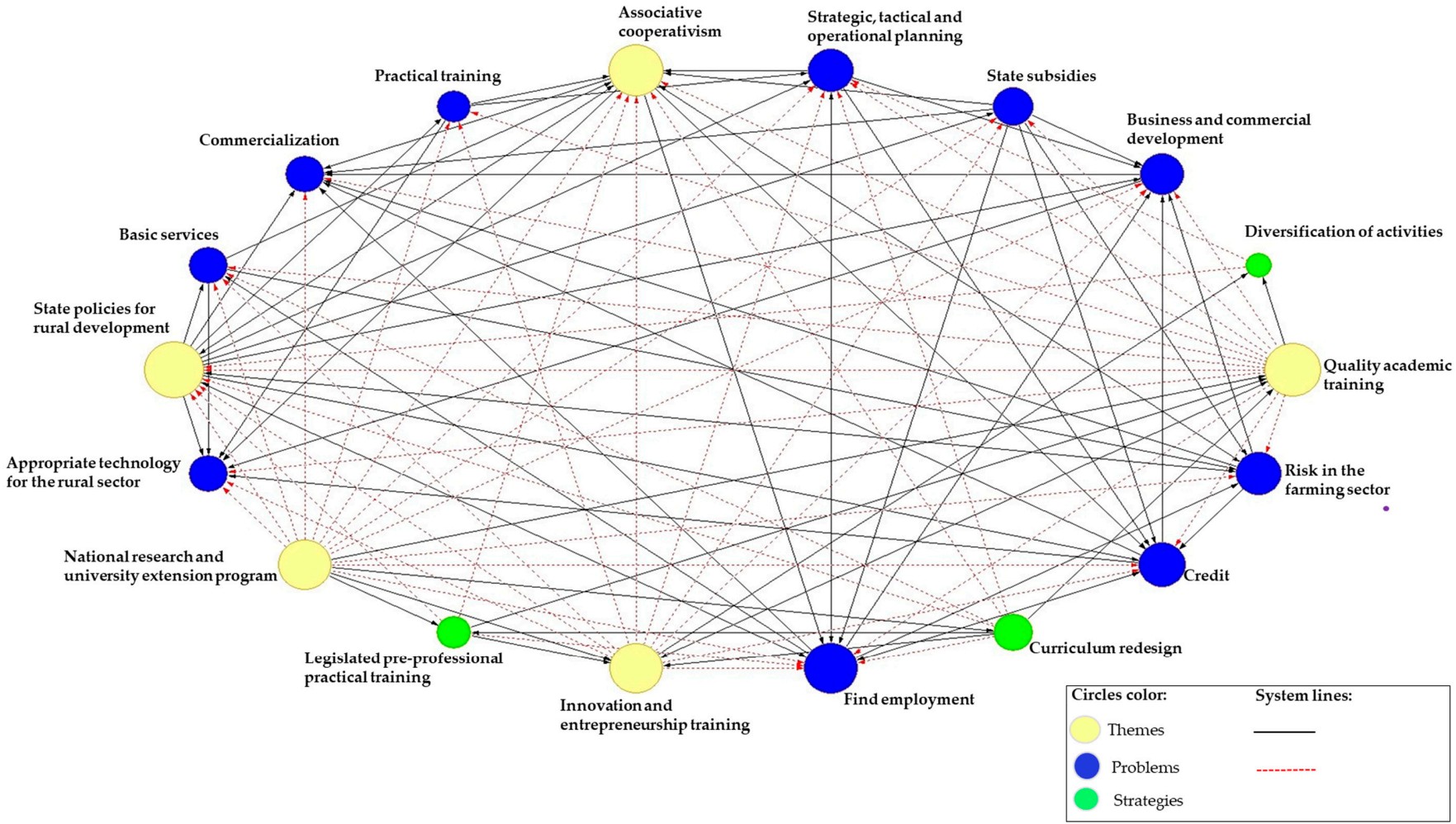

**Figure 2.** Aggregated community FCM map for the two research groups (ULs and APMs combined).

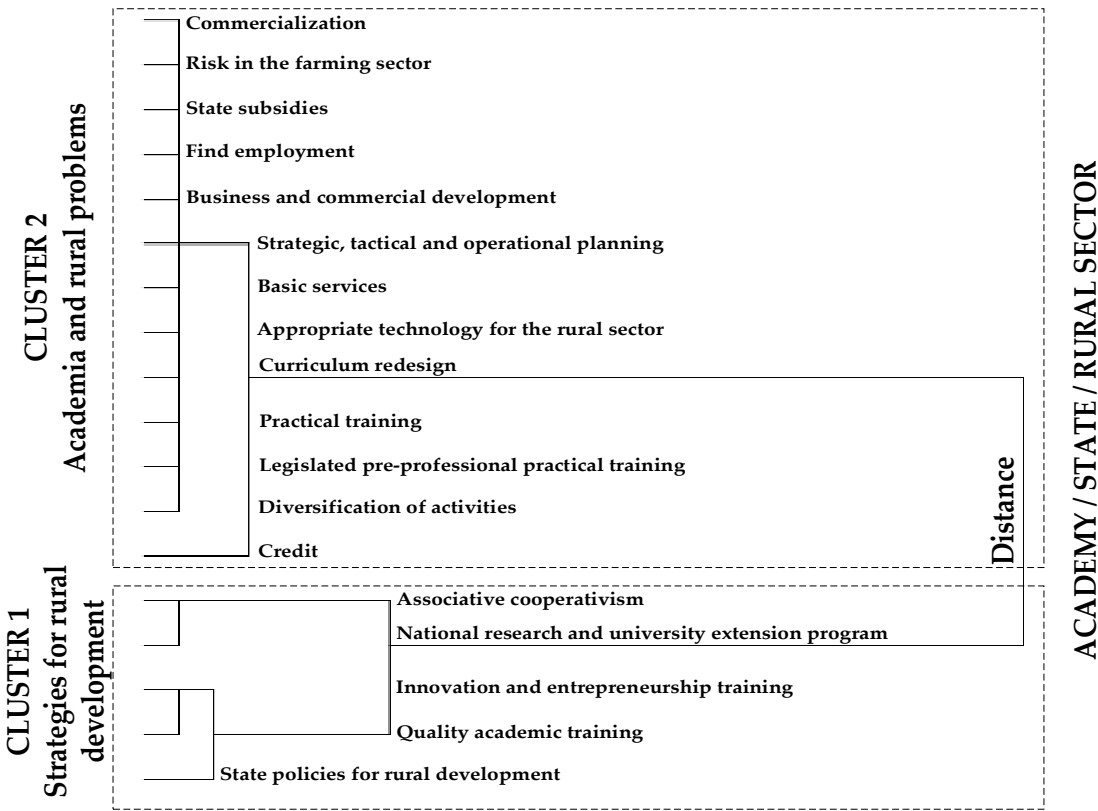

**Figure 3.** Dendrogram of the 18 categorized variables that represent the centrality values of the group that was interviewed (ULs and APMs).

## 4. Discussion

The two research groups (ULs and APMs) and the community group (38 interviewees) are in agreement on the importance and priority of the transmitter variable "National research and university extension program"—NRUEP (academia/state) and the receiver variable "Commercialization". This means that if academia and the state were to implement the NRUEP, they would be able to improve commercialization in the Ecuadorian agricultural sector, which in turn would drive agricultural graduates to return to the rural production sector. In effect, Kumar and Kumar [13], determine the importance of integrating research with the needs of the production sector, so that it can be part of the solution for the problems facing the agricultural sector; these being, reduction in productivity, lack of employment opportunities, and depletion of natural resources [19,20]. Furthermore, Quarouch et al. [48], Estévez [18], and Hidalgo et al. [15] determine that the rural production sector provides excellent opportunities for academia, the state, private companies, cooperation, organizations of producers, and the population to work together to diversify rural activities and generate non-agricultural employment in these areas. For example, in Latin American countries such as Uruguay, the innovation policy has been strengthened, creating a new institutionality with the ability to influence the allocation of public funds. The interaction between actors is carried out through national science, technology, and innovation plans, based on common objectives [49]. The results that have been obtained show the involvement of the state in allocating resources, and positive results have been achieved with scientific and research activities, as well as in companies. This has helped to perfect the political decision making process in this area, stimulating conversations and debates on innovation. Therefore, coordinated, collaborative, and cooperative relationships between universities and rural sectors are an essential part of the University Extension process, and will be of mutual benefit to all. Morales et al. [50] state that modern universities should integrate and connect University Extension programs with research and teaching, and these three functions are important for achieving excellence in academic training.

We should ask ourselves, how does the interaction between actors support the production sector? In response to this, Chiriboga [16] states that well-planned and joint projects can achieve successful results in terms of training, services, and information aimed at small and medium producers.

Furthermore, taking into account the importance of the FCM variables' centrality values for decision making, the ULs and APMs research groups' perceptions do not completely coincide with the five most important categorized variables (State Policies for Rural Development—SPRD, Quality Academic Training—QAT, Innovation and Entrepreneurship Training—IET, National Research and University Extension Program—NRUEP, and Associative Cooperativism—AC). The "credit" variable stated by the APMs groups which replaces the NRUEP variable, shows that academia and the state meet their responsibilities, although it is not perceived as being joined-up work. When considering the opinions of the overall group that was interviewed, with regards to agricultural graduates not returning to where they came from, it is determined that SPRD and QAT are strategies in which the state and academia can play a role by implementing research and university extension programs, to train competent agricultural professionals with innovation and entrepreneurship skills, thus driving associative work in rural areas. In contrast, the APM group believe that credit should be part of SPRD. Amongst other things, this will help to: (a) Avoid that small agricultural producers become involved in commercial processes that make them depend on 'chulqueros' (loan sharks), and fall into spiraling debt that makes them poor, and (b) encourage young people to see the agricultural sector as an investment opportunity and source of employment. This is a criteria that is also shared by Bednaríková et al. [20]. Similarly, Calabrò and Vieri [51] determine that the creation of sectoral and intersectoral state policies is an integral part of an official rural development strategy for building, supporting, and strengthening the agricultural sector, as well as all the areas and employment activities linked to the rural sector.

The agro-industry and agribusiness models that are developed in Ecuador in this era of globalization and debt crisis are persevering with the strengthening of production sectors that supports exports and agro-industry [52]. Similarly, the rural environment has been invaded by mining and flower-growing businesses, who are disrupting sustainable production in the rural environment. As a result, academia should debate these topics in order not to give into market demands and target support at creating policies that help small and medium producers. In this context, the transition of higher education to the labor market of the agricultural graduates make ourselves ask: Why do young graduates not return to the rural sector? Rerat [19] established that big cities offer more opportunities to highly educated individuals. The results identified 18 categorized variables as factors that affect the choice of the agricultural graduates not to return to the rural sector. In this regard, Fullerton [53], states that it is important for universities to be involved with rural communities so that they can improve and guarantee academic training with practical experiences and knowledge of the area, thus driving innovation and entrepreneurship in the rural sector. Similarly, Mojarradi and Karamidehkordi [54] state that contemplating theoretical training with experience and knowledge of the context will encourage graduates to return to the rural sector, as they will be responsible for the production of food in the near future.

The average centrality value of 1.77 of 5 grouped variables which correspond to Strategies for Rural Development—SFRD, implies that institutions with agricultural development capabilities should start working together to support rural development and achieve common objectives. The average centrality value of 0.41 of 13 variables grouped as Academia and Rural Problems—ARP, does not necessarily imply that these are less important than those grouped as SFRD. However, the highly significant difference ($p = 4.03e\text{-}08$) between the two groups of variables (SFRD and ARP), highlights the need to prioritize the implementation of SFRD, with policies directed at the rural sector and quality academic training in order to meet the needs of the rural production sector. Machado and Salgado [55] therefore state that academic work to support the design of policies in terms of effects and impacts has already been analyzed as a case study. Ramírez and García [56] highlight how joint University–Company–State projects help achieve management, entrepreneurship, and project innovation initiatives by working together. However, Lacki [57] states that if the support and intervention for the development of the rural

production sector is left as the sole responsibility of the state, rural products would be immobilized and dependent on external help, without trying to help themselves, combine their collective knowledge, and work together. Practical work should help to understand the realities facing an area, so that students are capable of establishing dialogues with the population, with actors from the production system, with business owners, enabling them to come up with their own proposals which are aligned to the reality of the different territories.

The work carried out could be complemented by future studies on the analysis of factors linked to a more personal sphere among graduates, and which could encompass the vision of this complex problem.

## 5. Conclusions

The use of fuzzy cognitive maps and hierarchical clusters as analysis tools enabled a practical diagnosis of a complex and highly interrelated reality, particularly between academia and the rural production sector. In this study, we have considered two researcher groups (ULs and APMs), whose perspectives allowed us to understand the critical knowledge position of each group of agricultural graduates regarding the decision not to return. However, the community map showed a general view as a whole, which helps us to consolidate a general outlook of what is happening with the agricultural graduates in Ecuador. The results of this study have led to the identification of five strategic variables that influence the phenomenon of agricultural graduates not returning to the rural sector in Ecuador. These categorized variables are: State policies for rural development, quality academic training, innovation and entrepreneurship training, a national research and university extension program, and finally, a drive towards associative cooperativism. Likewise, the study identified ten problems that affect the non-return of the agricultural graduate to the rural sector and three strategies where the participation of the government, the academy, and the rural sector is crucial. Public policies should be formed around these variables to improve living conditions and production in the rural areas, leading to the emergence of a diversified and sustainable economy and the creation of jobs and income streams. Universities, along with local governments in Ecuador, should promote local experimentation, technological innovation, and research, primarily in the agro-production sector, leading to the creation of local learning and innovation networks.

The identification of these priorities could steer state authorities and academia to aim their efforts at encouraging agricultural graduates to work in the rural production sector. From this specific point of view of academic training, the inclusion of programs to connect with society as well as research involves responding to the needs of the environment. In Ecuador, there is a lack of experience with these connection projects that help to test and validate strategies in situ, which will help to take on the challenge of bringing the university closer to the rural production sector, in order to respond to the expectations and demands relating to academic training amongst committed agricultural professionals.

**Author Contributions:** Conception of the paper, all authors; data acquisition, B.B.C.; data analysis and writing of original draft, B.B.C. and C.J.; data analysis, and methodological framework, M.S. and B.B.C.; supervision, writing—review and editing, and funding acquisition, B.B.C. and J.L.Y.B. All authors have read and agree to the published version of the manuscript.

**Funding:** This research received no external funding.

**Acknowledgments:** The authors grateful for the support of the Secretary of Higher Education, Science, Technology, and Innovation (SENESCYT-Ecuador). To agricultural planning professors and managers who provided information for this study.

**Conflicts of Interest:** The authors declare no conflict of interest.

## Appendix A

Definition of the categorized variables.

| No | Categorized Variables (Initial Format) | Definition |
|---|---|---|
| 1. | State policies for the development of the rural sector | Strategic guidelines to stimulate rural development, to promote sustainable agricultural production. |
| 2. | Basic services that are lacking in the rural sector | Relating to education, health, housing, communication, drinking water, and sewage services in the rural sector. |
| 3. | Problems with commercializing agricultural products | The supply and demand of agricultural products with price restrictions. A fair price is not paid to producers. |
| 4. | Inadequate practical training of agricultural professionals | Encouraging practical work in the countryside as part of the academic training for agricultural graduates. |
| 5. | Low levels of associative cooperativism between farmers | Create interinstitutional work, to improve associative cooperativism amongst farmers and agricultural graduates, ensuring gender inclusion. |
| 6. | Inadequate strategic, tactical, and operational planning in the rural farming sector | Establish a strategic agricultural plan in the short, medium, and long term that responds to the needs and potentiality of the rural production sector. |
| 7. | Insufficient subsidies for farmers in the rural sector | Support from the state for small and medium agricultural producers. |
| 8. | Insufficient business and commercial development in the rural sector | Driving the development of rural agribusiness, to add value to raw materials. |
| 9. | Diversification of activities in the rural sector | Sustainable exploitation of existing resources in rural areas (cultural tourism, services, development of rural agribusiness, provision of services, etc.) |
| 10. | Quality academic training | A young, well-trained workforce with a competitive advantage, that responds to society's needs, with values that respect culture and traditions. |
| 11. | Risk in the farming sector | Factors such as weather, illnesses or plagues, and theft, which can cause a decrease in, or loss of production. |
| 12. | Difficulty accessing credit | Reduce institutional criteria for guaranteeing credit, with differentiated and accessible interest rates for rural producers. |
| 13. | Redesign of the university agricultural curriculum | Adjust the study plans based on the need of the rural agricultural sectors. |
| 14. | Agricultural professionals trained to secure a job rather than start a business | A paradigm shift in the academic training of agricultural professionals, so that they can create jobs. |
| 15. | Promote innovation and entrepreneurship in academic training | Drive creativity, innovation, and entrepreneurship within the academic training of agricultural professionals. |
| 16. | Reorganization of legislated pre-professional practical training in the rural sector | Establish a model for managing pre-professional training linked to the rural production sector. |
| 17. | Create a national research and university extension program | Help with problem solving, exploit opportunities, and meet the needs facing the rural sector. |
| 18. | Insufficient availability of appropriate technology in the rural sector | Experiment with primary production and rural agribusiness using environmentally friendly technology. |

## Appendix B

Identification and categorization of variables for the interviewees.

| Categorized Variables (Initial Format) | Variables | Weight |
|---|---|---|
| State policies for the development of the rural sector | State policies do not strengthen the rural sector | + |
| | Policies that facilitate credit are not established | + |
| | There are no policies that encourage young people to work in the rural sector | + |
| | State policies do not guarantee the commercialization of agricultural products | + |
| | There are no state policies for the rural sector | + |
| | There are no incentive policies for strengthening the rural sector | + |
| | The rural sector is not a priority for the governments in office | + |
| | Specific policies for the rural sector are not created | + |
| | Poor implementation of policies that benefit the rural sector | + |
| | State policies do not guarantee economic stability in the rural sector | + |
| | State policies favor urban planning based on the services it offers | + |
| | Weak state policies that benefit small and medium producers | + |
| | There are no policies that support agroecology | + |
| | There are no policies that promote rural development | + |
| | Interinstitutional collaboration as a state policy is weak | + |
| | State policies that favor the academic sector are weak | + |
| | Poor strengthening of agroecological exhibitions as a state policy | + |
| | State policy does not promote adequate planning on the use of land | + |
| | State policy does not drive entrepreneurship in the rural sector | + |
| | State policy for the agricultural sector is not long-term | + |
| Difficulty accessing credit | Limited access to credit with low interest rates | + |
| | Limited access to credit | + |
| | Difficulty accessing credit for starting a production business | + |
| Problems with commercializing agricultural products | Inadequate policies for commercializing agricultural products | + |
| | The raw materials produced directly by the producer are not sold at fair prices | + |
| | High risk with farming due to instability of prices for agricultural products | + |
| | The state does not help to drive commercialization | + |
| | Farming losses due to commercialization problems | + |
| | Inadequate commercialization systems | + |
| Basic services that are lacking in the rural sector | Inadequate basic services in the rural sector compared to the urban sector | + |
| | Public policies favor the urban sector, driving migration from the countryside to towns | + |
| | Poor technological services in the rural sector | + |
| | Poor quality services in the rural sector | + |
| | The road system in the rural sector is in poor condition | + |
| Low levels of associative cooperativism between farmers | Poor university training with an associative focus | + |
| | Poor organization amongst producers | + |
| | Poor organizational strength | + |
| | Lack of interinstitutional associativity | + |
| | Lack of associativity amongst young people in order to improve entrepreneurship in the rural sector | + |
| | Lack of organization amongst producers including young people | + |
| Insufficient availability of appropriate technology in the rural sector | Limited modernization in the rural sector to add value to raw materials | + |
| | Low level of environmentally friendly technological development in the rural sector | + |
| | Technology that can strengthen the rural sector is not developed. So their culture does not change | + |
| | It is necessary to ensure technology does not cause environmental damage in the rural sector | + |
| | It is necessary to develop new technologies in the rural sector that respond to climate change | + |

| Categorized Variables (Initial Format) | Variables | Weight |
|---|---|---|
| Insufficient subsidies for farmers in the rural sector | Lack of subsidies for sustainable agricultural production | + |
| | There are no subsidies for small and medium farmers | + |
| | Establish subsidies as incentives to strengthen agricultural production | + |
| | There are no subsidies that promote clean production | + |
| | There are no subsidies as part of state policy, as farming is considered high risk | + |
| Inadequate strategic tactical and operational planning in the rural farming sector | There is a lack of adequate agricultural planning | + |
| | There is no agricultural planning relating to the use of land | + |
| | There is no long-term agricultural planning | + |
| Insufficient business and commercial development in the rural sector | Poor links between universities and companies | + |
| | Low levels of development amongst companies for hiring agricultural professionals | + |
| | Low levels of development of businesses dedicated to the agricultural sector | + |
| | Weak presence of companies that market agricultural products | + |
| | The creation of micro-enterprises is not promoted in the rural sector | + |
| | Production activities in the rural sector do not promote the development of micro-enterprises | + |
| Risk in the agricultural sector | High risk in agricultural production | + |
| | Agriculture is high risk, due to weather conditions and product prices | + |
| | Eroded land and low yields with agricultural production | + |
| | Poor profitability of agricultural production | + |
| | The prices of agricultural products are not competitive | + |
| | Farming is risky due to the instability of product prices | + |
| Poor quality practical training of agricultural engineers | University training is not directly linked to the producers | + |
| | University training is more theoretical than practical | + |
| | Poor links between the university and the rural sector as part of university training | + |
| | University training does not respond to the producers' interdisciplinary, transdisciplinary, and multidisciplinary work. | + |
| Professionals trained just to find employment | On the whole, university professionals seek employment | + |
| | University training is generally in order to find a job | + |
| | Agricultural professionals seek employment because they think farming is not profitable | + |
| | There are a lack of agricultural professionals, and in reality, there are a lack of job positions | + |
| | Parents who have small and medium farming businesses encourage their children to leave the rural sector as they do not have arable land | + |
| Diversification of activities in the rural sector | Promote new activities in the rural sector, such as tourism and agro-industry | − |
| | Provide incentives in order to stimulate the economy in the rural sector | − |
| | Strengthen agroecology in the rural sector | − |
| | Provide added value to raw materials and create micro-enterprises in the rural sector | − |
| | Develop aquaculture activities in the rural sector, taking into account the potential of water resources and working in parallel to repair the land, especially in the mountains | − |
| | State policy for diversifying production in the rural sector | − |
| Quality academic training | Academic training should be more practical than theoretical | − |
| | Training with an entrepreneurial focus | − |
| | Improve interdisciplinary work in academic training | − |
| | Training based on experiences from other universities, as in the case of Earth | − |
| | Comprehensive academic training | − |
| | Training that is more closely linked to the rural sector | − |
| | Preserve ancestral knowledge | − |
| | Strengthen academic training in the farming field | − |
| | Comprehensive training of agricultural professionals | − |
| | Training with a social focus | − |

| Categorized Variables (Initial Format) | Variables | Weight |
|---|---|---|
| Redesign of the university agricultural curriculum | Agricultural degree curriculum maps based on innovation and entrepreneurship | – |
| | Agricultural degree curriculum maps that respond to the needs of the rural sector | – |
| | Reengineering of agricultural degree curriculum maps | – |
| | Strengthen curriculum maps with a social foundation | – |
| | Drive environmental sociology and community development within curriculum maps | – |
| | Curriculum maps with a focus on cultural preservation | – |
| | Curriculum maps that provide more practical training | – |
| Promote innovation and entrepreneurship in academic training | Improve interdisciplinary work in order to improve entrepreneurship | – |
| | Strengthen academic training based on entrepreneurship | – |
| | Improve academic training with a focus on innovation and entrepreneurship | – |
| Reorganization of legislated pre-professional practical training in the rural sector | Link pre-professional practical training to the rural sector | – |
| | Strengthen pre-professional practical training linked to the rural sector | – |
| | Involve students so that they understand the problems facing the rural sector through pre-professional practical training | – |
| | Pre-professional practical training should be carried out at the place of origin, in the case of students who come from the rural sector | – |
| Create a national research and university extension program | Strengthen the link between the community and academia | – |
| | Improve research and links with the community | – |
| | Differentiate university extension from rural extension | – |
| | Create a national system for research and university extension | – |
| | University extension should be strengthened so that future professionals return to this space to improve their knowledge | – |
| | Improve research that responds to the needs of the rural sector | – |
| | Connect academia with the production sector | – |
| | Bring research and the link between university and society together | – |
| | Incorporate research, the link between university and society, and academic training | – |
| | Greater investment aimed at adding value to raw materials | – |
| | Connect farmers with research | – |
| | Research should be considered with the farmers | – |
| | Research should be based on entrepreneurship | – |
| | Develop research, create knowledge, and apply it | – |

## Appendix C

**Table A1.** Centrality of the FCM for the Categorized Variables in the 2 research groups and community group (ULs and APMs).

| Categorized Variables | Centrality | | |
|---|---|---|---|
| | ULs | APMs | ULs & APMs |
| Quality academic training | 1.91 | 2.13 | 2.04 |
| State policies for rural development | 1.76 | 2.62 | 2.28 |
| Innovation and entrepreneurship training | 1.71 | 1.90 | 1.82 |
| National research and university extension program | 1.61 | 1.00 | 1.24 |
| Associative cooperativism | 1.58 | 1.38 | 1.46 |
| Curriculum redesign | 0.80 | - | 0.32 |
| Find employment | 0.66 | 0.51 | 0.57 |
| Strategic, tactical, and operational planning | 0.67 | 0.39 | 0.50 |
| Legislated pre-professional practical training | 0.63 | - | 0.25 |
| State subsidies | 0.57 | 0.59 | 0.58 |
| Risk in the farming sector | 0.49 | 0.36 | 0.41 |
| Credit | 0.49 | 1.01 | 0.80 |
| Commercialization | 0.28 | 0.50 | 0.41 |
| Basic services | 0.23 | 0.35 | 0.30 |
| Practical training | - | 0.33 | 0.20 |
| Business and commercial development | - | 0.91 | 0.55 |
| Diversification of activities | - | 0.22 | 0.13 |
| Appropriate technology for the rural sector | - | 0.49 | 0.30 |

ULs = University Lecturers; GPAs = Agricultural Planning Managers; ULs & APMs = Community Group.

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
