# Peer review of "Identification of Variables That Cause Agricultural Graduates Not to Return to the Rural Sector in Ecuador. Application of Fuzzy Cognitive Maps"

_sustainability, doi:10.3390/su12104270_

Round 1
Reviewer 1 Report
The article presented meets the requirements to be published, responds to research problems from qualitative research. It is well explained and systematized, it would only include a table with the categorical system and their definitions as a means to systematize and explain in a more correct way the research methodology used. The results and conclusions are related to the research questions asked.
Author Response
Point 1: The article presented meets the requirements to be published, responds to research problems from qualitative research. It is well explained and systematized, it would only include a table with the categorical system and their definitions as a means to systematize and explain in a more correct way the research methodology used. The results and conclusions are related to the research questions asked.
Response 1: According with your suggestions it has added a table called “Definition of the categorized variables”, which is showed in the Appendix A, and it is called in the lines 431-432

Reviewer 2 Report
Dear authors,
This is a good paper. For its even better improvement I have the following suggestions:
- Lines 46-50: Re-phrase: after you say that in US graduates return to where they came from, the next sentences (that seems like a following conclusion) contradicts this.
- Line 81, source 19 and line90 source 23 name the country fir which the results are presented
- line 81 16.4m go to 16.4 mil.
- better justify when/why you use the two categories of experts to document behaviour of graduates
- the paper does not specifically/explicitly comment on the reasons for which agricultural graduates do not return in the rural areas. I suggest to change the title to better reflect the content in something like (see lines 144-145): Variables influencing the relationship between agricultural graduates and rural areas. And complete under the discussion and conclusion sections where factors cause graduates not to return to rural areas.
- Lines 167-170: clarify the concepts degree and postgraduate degree: you mean Bachelor degree and other postgraduate degrees? you mean degree (any type) and postgraduate training? How the inclusion of this information affects the quality of your sample?
- Section 2.3.: explain how you have developed the a priori coding system. Also in text and in appendix A make a difference in terms if words between the 117 variables and the 18 categorized variables. Later in the text other words are used to describe the same : the 18 groups. Decide on a term (different from other terms in the research) and use it consistently over the paper.
- Line 203: "....within it." Explain/include who is "it".
- Section 2.4.1., lines 217-226:better explain the complexity index AND include in the explanation of the level of participation of variables in the system, the interpretation as well: what does it actually mean that (ex:) receiver variables receive connections with positive weights from other variable? How do we interpret? What is the actual relationship between these variables? The same for all other variables explained here.
- Section 3.3. Better explain the disagreement of the two groups on the priority of variables (by mean of example)
- Section 3.5. Explain the high significant differences between the two clusters
- In figure 3, the names of the clusters is NOT translated in English.
Author Response
Point 1. Lines 46-50: Re-phrase: after you say that in US graduates return to where they came from, the next sentences (that seems like a following conclusion) contradicts this.
Response 1: Line 46-50: According with your suggestions it has made the suggested change, and it has deleted the sentence “The low likelihood of youngsters remaining in the countryside has become a common characteristic, both in developed countries as well as developing countries[8]”.
Point 2. Line 81, source 19 and line90 source 23 name the country fir which the results are presented.
Response 2 : Line 81: It has added the country where the data proceed, which was Ecuador. Line 89-90: It has added the country where the data proceed, which was Castilla de León-Spain.
Point 3. line 81 16.4m go to 16.4 mil.
Response 3: Line 81-82 : It has changed to the suggested numeric format.
Point 4. better justify when/why you use the two categories of experts to document behaviour of graduates.
Response 4: The paper has not set out to study those reasons that graduates do not return which are at a more personal or micro level. Although this would be a complementary and interesting addition, the paper aims to understand the macro and overall reasons. This explains, logically, why the groups of experts that were interviewed are primarily university professors and agricultural planning managers.
We added these texts: Lines 162-164. It has added in the text “The interviews were divided in two research groups: academic and agricultural planning sector. The main purpose to establish two categories was to analyze perspectives from academic vision, and an integrated management vision”
Lines 399-403. Also, it has highlighted the use of categories in the conclusion with the text: “In this study, we have considered two researcher groups (ULs and APMs), which perspective allowed us understand the critical knowledge position of each group regard not to return of agricultural graduates. However, the community map showed a general view as a whole, which help us to consolidate a general outlook of what is happening with the agricultural graduates in Ecuador.”
Point 5. The paper does not specifically/explicitly comment on the reasons for which agricultural graduates do not return in the rural areas. I suggest to change the title to better reflect the content in something like (see lines 144-145): Variables influencing the relationship between agricultural graduates and rural areas. And complete under the discussion and conclusion sections where factors cause graduates not to return to rural areas.
Response 5: In terms of the change of title that the reviewer suggests, we would prefer to maintain the original one as we think it better reflects the focus that we wanted to give this study. However, we clearly understand your advice and we want to try to clarify this.
The paper has not set out to study those reasons that graduates do not return which are at a more personal or micro level. Although this would be a complementary and interesting addition, the paper aims to understand the macro and overall reasons. This explains, logically, why the groups of experts that were interviewed are primarily university professors and agricultural planning managers, as also mentioned in the previous point.
The experts were explicitly asked about the perceptions on the causes/problems that explain that agricultural graduates are not returning to the rural sector. Therefore, the results steer us towards five strategic variables that are considered to be the most important, amongst academia, the state and the rural sector.
Having covered the predicament in general, and from line 99 of the introduction, macro aspects are reviewed which are specifically of interest in Ecuador. Before finishing this section, we mention the need for these scientific studies to help inform policies and, in general, decision makers. We therefore reiterate the focus on macro or external aspects.
In general, we understand the need to be more explicit and we have therefore incorporated the following sentences in the sections that you suggest:
Line 367-369: In the discussion section, we have included a phrase related to the topic of non-return of agricultural graduates.
Lines 399-403: In the conclusion section, we have included a sentence, which indicates the main result of the study, which highlighted the non-return of the agricultural graduates.
Lines 393-395 : The work carried out could be complemented by future studies on the analysis of factors linked to a more personal sphere among graduates, and which could encompass the vision of this complex problem.
Point 6. Lines 167-170: clarify the concepts degree and postgraduate degree: you mean Bachelor degree and other postgraduate degrees? you mean degree (any type) and postgraduate training? How the inclusion of this information affects the quality of your sample?
Response 6: Lines 172: It has changed “degree” to “Bachelor degree” Lines 173-174: In the case of the postgraduate degree is referred to master education level. Lines 174-176: It has highlighted the importance to have a group of experts from the academic vision.
Point 7. Section 2.3.: explain how you have developed the a priori coding system. Also in text and in appendix A make a difference in terms if words between the 117 variables and the 18 categorized variables. Later in the text other words are used to describe the same: the 18 groups. Decide on a term (different from other terms in the research) and use it consistently over the paper.
Response 7: Lines 178-190: It has made a modification in the text, which explain in an explicit way how was made the coding system. It has added a sentence where it has highlighted that the categorization process were made injoint with the authors. On the other hand, it has standardized the name of the variables in the whole document based on Table 2. Thus, the paper is showing the condensed format of the categorized variables.
Lines 191: Table 1
Lines 294: Figure 1
Lines 300: Figura 2
Lines 309: Figura 3
Línes 433: Appendix B
Línes 435: Appendix C
Lines 219, 247, 262, 286, 341. The word categorized was inserted
Point 8. Line 203: "....within it." Explain/include who is "it".
Response 8: Line 211-212: It has explained in the text what we referred with “it”.
Point 9. Section 2.4.1., lines 217-226: better explain the complexity index AND include in the explanation of the level of participation of variables in the system, the interpretation as well: what does it actually mean that (ex:) receiver variables receive connections with positive weights from other variable? How do we interpret? What is the actual relationship between these variables? The same for all other variables explained here.
Response 9: Line 233-242: It has explained in a better way the complexity index based on the research of Ozesmi and Ozesmi (2003). In addition, we have include the explanation of the level of participation based on the research presented by Eden et al. (1992).
Point 10. Section 3.3. Better explain the disagreement of the two groups on the priority of variables (by mean of example).
Response 10: Line 291-293. In section 3.3: It has added an example of how we have determined the disagreement of the two groups.
Point 11. Section 3.5. Explain the high significant differences between the two clusters.
Response 11: Lines 305-308. In section 3.5: It has added a brief explanation of what it means high significant difference. It is important to mention that if the value obtained is lower than p-value<0.05; it will mean that there are significant differences.
Point 12. In figure 3, the names of the clusters is NOT translated in English.
Response 12: Line 309. In figure 3: It has made the change suggested to “Strategies for rural development” y “Academy and rural problems”
